# Full-length human GLP-1 receptor structure without orthosteric ligands

Fan Wu 🄳 [1,2,3], Linlin Yang[4,15], Kaini Hang[1,2,3,15], Mette Laursen 🄳 [5,15], Lijie Wu[2], Gye Won Han[6], Qiansheng Ren[7], Nikolaj Kulahin Roed[5], Guangyao Lin[2,3], Michael A. Hanson[8], Hualiang Jiang[9,10,11], Ming-Wei Wang 🄳 [2,9,12,13], Steffen Reedtz-Runge 🄳 [5], Gaojie Song 🄳 [14✉] & Raymond C. Stevens[1,2,6✉]

Glucagon-like peptide-1 receptor (GLP-1R) is a class B G protein-coupled receptor that plays an important role in glucose homeostasis and treatment of type 2 diabetes. Structures of full-length class B receptors were determined in complex with their orthosteric agonist peptides, however, little is known about their extracellular domain (ECD) conformations in the absence of orthosteric ligands, which has limited our understanding of their activation mechanism. Here, we report the 3.2 Å resolution, peptide-free crystal structure of the full-length human GLP-1R in an inactive state, which reveals a unique closed conformation of the ECD. Disulfide cross-linking validates the physiological relevance of the closed conformation, while electron microscopy (EM) and molecular dynamic (MD) simulations suggest a large degree of conformational dynamics of ECD that is necessary for binding GLP-1. Our inactive structure represents a snapshot of the peptide-free GLP-1R and provides insights into the activation pathway of this receptor family.

[1] iHuman Institute, ShanghaiTech University, Shanghai, China. [2] School of Life Science and Technology, ShanghaiTech University, Shanghai, China. [3] University of Chinese Academy of Sciences, Beijing, China. [4] Department of Pharmacology, School of Basic Medical Sciences, Zhengzhou University, Zhengzhou, China. [5] Novo Nordisk A/S, Novo Nordisk Park, Copenhagen, Denmark. [6] Departments of Biological Sciences and Chemistry, Bridge Institute, USC Michelson Center for Convergent Bioscience, University of Southern California, Los Angeles, California, USA. [7] Novo Nordisk Research Center, Beijing, China. [8] GPCR Consortium, San Marcos, California, USA. [9] CAS Key Laboratory of Receptor Research, Shanghai Institute of Materia Medica, Chinese Academy of Sciences, Shanghai, China. [10] State Key Laboratory of Drug Research, Shanghai Institute of Materia Medica, Chinese Academy of Sciences, Shanghai, China. [11] Drug Discovery and Design Center, Shanghai Institute of Materia Medica, Chinese Academy of Sciences, Shanghai, China. [12] The National Center for Drug Screening, Shanghai Institute of Materia Medica, Chinese Academy of Sciences, Shanghai, China. [13] School of Pharmacy, Fudan University, Shanghai, China. [14] Shanghai Key Laboratory of Regulatory Biology, Institute of Biomedical Sciences and School of Life Sciences, East China Normal University, Shanghai, China. [15] These authors contributed equally: Linlin Yang, Kaini Hang, Mette Laursen. ✉email: gjsong@bio.ecnu.edu.cn; stevens@shanghaitech.edu.cn

C lass B G protein-coupled receptors (GPCRs), whose endogenous ligands are peptide hormones, are key mediators of normal human physiology and serve as valuable drug targets for many diseases including diabetes, metabolic syndrome, osteoporosis, migraine, depression, and anxiety[1]. They include an N-terminal 120–160 residue extracellular domain (ECD) and a C-terminal transmembrane domain (TMD), both of which are important for peptide hormone binding and activation[2]. The two-domain binding mode suggests that the C-terminus of the peptide hormone initiates recognition with the ECD. This initial recognition step allows the peptide's N-terminus to engage deep within the receptor TMD core, triggering a conformational change proximal to the intracellular region that results in G protein coupling and activation of the downstream signalling cascade[3–7]. Recent crystal and cryo-EM structures of class B GPCRs have revealed a relatively conserved binding mode of the peptide hormones and similar orientations between the ECD and TMD when fully activated[3–5,8,9].

Glucagon-like peptide-1 (GLP-1) is a key incretin hormone secreted in response to food intake[10]. GLP-1 acts on the GLP-1 receptor (GLP-1R) to lower blood glucose through enhanced glucose-dependent secretion of insulin, inhibition of glucagon secretion and slowed gastric emptying. It lowers body weight through reduced food intake. Therefore, peptide analogs of GLP-1 have been developed to treat type 2 diabetes and obesity, with the beneficial outcome of lowering cardiovascular risks[10,11]. Thus far, the binding poses of two full-length peptide-bound, active GLP-1R structures[3,4] and an active-like structure with a truncated peptide agonist (peptide 5)[12] have been detailed, as well as variable ligand-dependent ECD receptor conformations. Previous inactive GLP-1R TMD structures revealed how allosteric modulators can precisely regulate its function from the extra-helical binding sites[13]; however, the ECDs in these structures had been truncated, and hence the conformation of the ECD in peptide-free or inactive states remain unknown. Structures of the glucagon receptor (GCGR), the closest homolog of GLP-1R, have revealed key differences between the peptide-free and peptide-bound states, including an interesting β-sheet motif in the peptide-free structure comprised of the stalk region and extracellular loop 1 (ECL1)[9,14]. Notably, the stalk region and ECL1 are known to be important regulators for peptide hormone recognition and conformational dynamics[15,16]; however, their sequences are diversified within class B GPCRs. To evaluate if the peptide-free inactive state structure of GCGR is conserved within class B receptors, we determine the structure of full-length GLP-1R in the absence of its orthosteric peptide agonist. The structure reveals a closed conformation in which the peptide-binding region of the ECD interacts with the extracellular regions of the TMD. We verify the physiological relevance of this unique closed conformation by disulfide crosslinking experiments. Furthermore, results from negative stain electron microscopy (EM) and molecular dynamic (MD) simulations suggest conformational dynamics of GLP-1R during binding to GLP-1. The closed conformation of the peptide-free GLP-1R compared to GCGR is quite divergent in the extracellular region and may be a consequence of sequence diversity, which is important for understanding the signalling pathway for different receptors within the class B family.

## Results

**The inactive full-length GLP-1R structure.** The crystallization construct of the full-length human GLP-1R includes all of the thermostabilizing mutations present in the previous inactive GLP-1R TMD structures[13] (Methods, Supplementary Fig. 1). In addition, a non-competitive ECD-binding antibody, Fab fragment (Fab7F38)[17], was added along with the TMD-binding negative allosteric modulator (NAM) PF-06372222 (originally designed for

GCGR)[18] for co-crystallization and successful determination of the structure to 3.2 Å resolution (Fig. 1a, Supplementary Fig. 2). The TMD in the full-length structure shares a similar conformation with the previous inactive TMD structure with a $C_\alpha$ root mean squared deviation (r.m.s.d.) of 0.6 Å; the most significant structural differences were observed in the extracellular regions (Fig. 1b). In particular, ECL1 and ECL3, which were disordered in the previous TMD structure, are now ordered and form a α-helical conformation reminiscent of the peptide-bound GLP-1R structures. The ECD assumes a unique inactive conformation and interacts with Fab7F38 through the βA and βB strands as well as the L1 and L4 loops (Fig. 1a). The antibody epitope of the ECD does not overlap with the peptide-binding site, consistent with the non-competitive nature of the antibody in the cAMP assay (Fig. 1c). The antibody Fab7F38 appears to function by providing enhanced soluble surface area for crystal lattice packing, and indeed Fab7F38-bound GLP-1R can also assume an active conformation as shown by EM studies (see below), suggesting that Fab7F38 does not interfere with the conformational flexibility of the ECD during crystallization.

The orientation between the ECD and TMD in the GLP-1R–Fab7F38 structure differs markedly from that of previous GLP-1R full-length structures (Fig. 2a). In previous GLP-1- or exendin-P5-bound active structures[3,4], the ECD shows a fully extended open conformation, whereas in the peptide 5-bound active-like structure[12], the ECD is less extended since it makes fewer interactions with the truncated peptide 5. Remarkably, in our GLP-1R–Fab7F38 structure, since no peptide ligand is bound to the orthosteric pocket, the peptide-binding groove of the ECD is juxtaposed with the TMD interacting with ECL1 and ECL3. Specifically, in the GLP-1R–Fab7F38 structure, the tip of the ECD (measured at the $C_\alpha$ of A57) moves by 18 Å and 28 Å from their positions in the peptide 5- and GLP-1- bound structures, respectively (Fig. 2a).

Within the TMD, we observed large structural shifts compared to the active peptide-bound structures, particularly in the extracellular halves of the TMD in the GLP-1R–Fab7F38 structure. Compared to the active state, ECL1 moves toward helix I by 5 Å when measured at the N-terminal tip of the helix ($C_\alpha$ of Q211), and ECL1 residue W214 is reoriented ~180° from an outside-facing position to a position pointing towards helix I. Furthermore, the α-helical ECL3 moves toward the TMD core by 12.3 Å in the inactive GLP-1R structure (measured at $C_\alpha$ of T378); conversely, the stalk and the extracellular half of helix II move 10–12 Å away from the orthosteric pocket (Fig. 2b). Despite these large structural differences, the TMD pocket volume of the GLP-1R–Fab7F38 structure (893 Å$^3$) is of similar scale as the ligand-occupied GLP-1R pockets (5VAI: 1036 Å$^3$; 5NX2: 883 Å$^3$), indicating that activation of the TMD occurs through reorganization of the helix bundle rather than the dramatic expansion or shrinkage of the binding pocket. The closing of the extracellular regions by the ECD and the reorganization of the TMD conformation interfere with the binding of GLP-1 to the orthosteric pocket in the inactive state of full-length GLP-1R (Supplementary Fig. 3).

**Inactive state stabilized by the ECD and ECL1/3.** In the GLP-1R–Fab7F38 structure, the saddle-like ECD loop S116-E127 covers the ECL1 helical region Q211-S219 that partially occupies the orthosteric ECD-binding site (Fig. 3a). On the other side, the αA of ECD (residues W33-R40) runs anti-parallel with the ECL3 (residues L379-T386) (Fig. 3b). The ECD–TMD interface is relatively small with a buried solvent-accessible surface of 1015 Å$^2$, in contrast to a total of 1632 Å$^2$ and 3278 Å$^2$ in the peptide 5-GLP-1R and GLP-1-GLP-1R interfaces, respectively. Moreover, although most of the residues in the ECD–TMD interface are hydrophilic, we did not observe any hydrogen bond

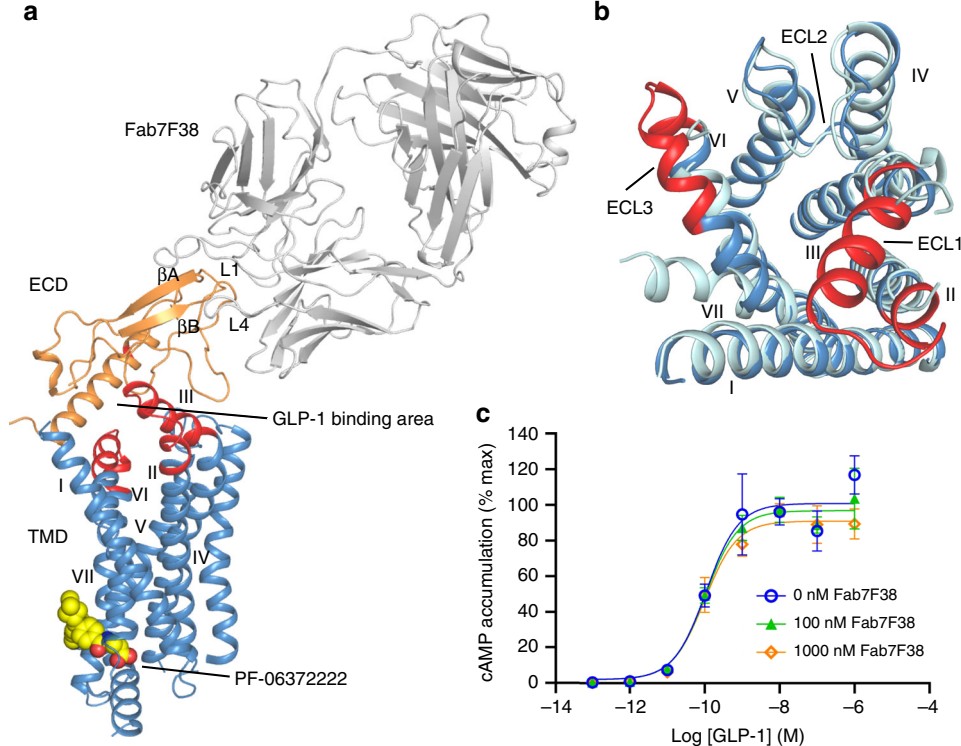

**Fig. 1 Overall structure of the GLP-1R–Fab7F38 complex. a** Cartoon representation of GLP-1R–Fab7F38 complex. Fab, ECD (residues R24-E127) and TMD (residues E128-F257 and E262-L422) are colored grey, orange and blue, respectively. The ECL1 (K202-S223) and ECL3 (F369-K383) residues that interact with the ECD are shown in red. Negative allosteric modulator (NAM; PF-06372222) is shown as spheres with yellow carbons. **b** Superposition of the full-length GLP-1R with previous TMD structure (cyan, PDB: 5VEW). Different regions are highlighted and labelled. **c** cAMP results show that Fab addition has no effect on GLP-1R activity. Data shown are mean ± s.e.m. of three independent experiments conducted in duplicate. Source data are provided as a Source Data file.

interactions between the two domains. To validate the observed closed conformation and to study their effects on receptor activation, we engineered disulfide bonds on the wild-type GLP-1R to lock the interactions between the ECD and ECL1/3. The results showed that GLP-1R mutant E127$^{ECD}$C–Q211$^{ECL1}$C substantially decreased the potency (~100 times) of GLP-1 in the cAMP accumulation assay, which could be recovered by adding 1 mM of dithiothreitol (DTT) (Fig. 3c, Supplementary Fig. 4). Likewise, the potency of the Q37$^{ECD}$C–L379$^{ECL3}$C mutant was also substantially compromised and could be reversed with DTT (Fig. 3d, Supplementary Fig. 4). These data indicate that the disulfide bonds of both E127$^{ECD}$C–Q211$^{ECL1}$C and Q37$^{ECD}$C–L379$^{ECL3}$C are formed on the native protein in the absence of the antibody, strengthening the hypothesis that the current structure represents a physiologically relevant inactive conformation of GLP-1R. The fact that the double-cysteine mutations only partially abolish the physiological function may imply that not all the expressed mutants have an intact disulfide bridge and that the observed conformation is only one possible inactive state on the cell surface, as supported by the EM data and MD simulations below.

**EM analysis of Fab7F38-bound GLP-1R**. The ECD of agonist-bound GLP-1R is known to assume different orientations in a ligand-dependent manner[3,4,12]. Conformational flexibility of the ECD was also observed in the Fab7F38-bound GLP-1R using negative stain EM single-particle analysis of a complex consisting of Fab7F38, semaglutide (a closely related analog of GLP-1 and approved drug for treatment of type 2 diabetes), detergent solubilized GLP-1R, Gs (nucleotide free), and Nb35 (Fig. 4, Supplementary Fig. 5). The 2D class averages clearly show

Fab7F38 bound to the ECD and the TMD in detergent micelle with the associated Gs protein (stabilized by Nb35). Interestingly, the 2D class averages reveal multiple conformations of the Fab7F38-bound ECD with some conformations closely resembling an open conformation as observed in the fully active structures (Fig. 4a). The ECD was reported to be relatively dynamic even in the presence of hormone peptide. In fact, in an analogous class B receptor, the parathyroid hormone receptor-1 (PTH1R), the ECD can adopt more than one conformation while bound to a long-acting PTH analog and in the Gs-coupled state[5]. Similarly, a preparation of calcitonin receptor (CTR)-calcitonin-Gs complex did not resolve clear density for the ECD of CTR attributed to partial flexibility[8]. Figure 4b shows a model of the Fab7F38-bound active state GLP-1–GLP-1R–Gs superimposed on a pair of 2D class averages, suggesting that GLP-1R can assume an open active conformation while bound to Fab7F38. In contrast, the active model does not align with the tilted 2D classes (Supplementary Fig. 5c). These conformations indicate flexibility of the ECD even in the Gs-bound state; however, the limited resolution of the EM data does not allow for a precise conclusion about the molecular details of the various conformations.

**MD simulations of apo GLP-1R**. MD simulations have been used to study the conformational dynamics of class B GPCRs and have provided valuable information regarding the molecular basis of receptor dynamics and ECD transitions between the inactive and active states[15,19]. To investigate the dynamics of the ECD toward TMD, we performed three independent 1-μs simulations based on the inactive full-length GLP-1R (Fig. 5, Supplementary Movies 1–3). Throughout the duration of the simulations, ECL1

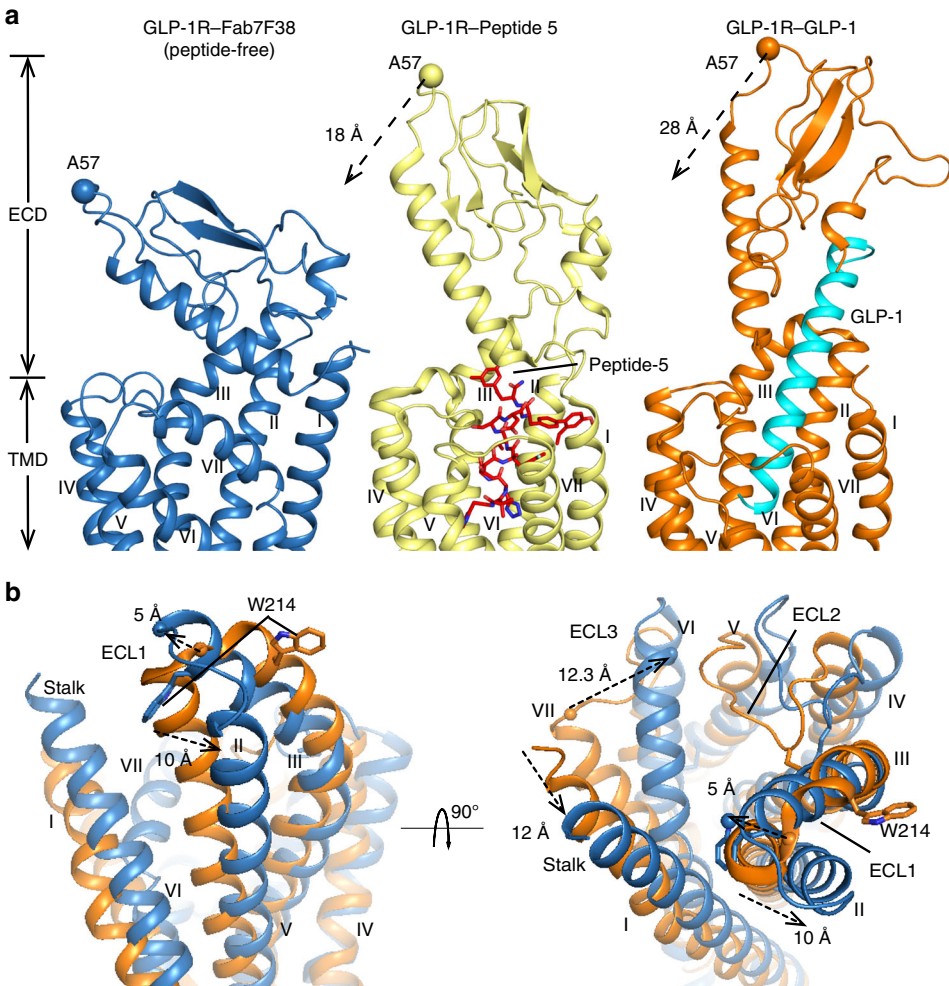

**Fig. 2 The inactive conformation of GLP-1R. a** Comparison of ECD orientation between the peptide-free GLP-1R–Fab7F38 structure (blue), the active-like peptide 5-bound GLP-1R structure (yellow; PDB: 5NX2), and the active GLP-1-bound GLP-1R–Gs complex structure (orange; PDB: 5VAI). Peptide 5 is shown as sticks with red carbons, GLP-1 is shown in cyan. **b** Superimposition of the TMDs between the GLP-1R–Fab7F38 (blue) and active GLP-1-bound GLP-1R–Gs complex (orange) structures. Significant changes are highlighted with dashed arrows. The reoriented residue (W214) in ECL1 is shown as sticks, landmark residues (A57$^{ECD}$, Q211$^{ECL1}$, T378$^{ECL3}$) for distance measurements are shown as spheres (C$_\alpha$).

and ECL3 maintained most of their original helical structure except for certain motions along with the adjacent transmembrane helices and the opposing ECD. We found that the orientations of ECD in two of the three trajectories (1 and 2) are relatively stable (average C$_\alpha$ r.m.s.d. 8.1 Å and 9.5 Å, respectively) because of the pre-existing restrictions by the interactions between the ECD and TMD (Fig. 5a–c). Notably, in the third trajectory, the ECD-TMD interactions are disrupted after 100 ns, the motions of the ECD are quite large in the range of 100–700 ns (the C$_\alpha$ r.m.s.d. is about 50 Å in the snapshot of 320 ns), and the molecule reaches a relatively stable conformation after 750 ns (Fig. 5a). Superposition of the third trajectory middle stage with the previous active GLP-1R structure (PDB: 5VAI) revealed the apo-state GLP-1R may adopt a similar extended ECD conformation as in the GLP-1-bound structure (Supplementary Fig. 6). Furthermore, in snapshots around 320–450 ns, the upper half of helix I moves together with the ECD away from the orthosteric pocket of the TMD, providing enough space for binding of the GLP-1's C-terminus to the ECD as well as docking of the N-terminus of GLP-1 to the orthosteric pocket in the TMD. Interestingly, in trajectory 3 after 750 ns, the dynamic ECD moves back toward the TMD and is stabilized in a conformation that forms contacts mainly with ECL1, resembling the closed

ECD conformation in the crystal structure. Generally, this ECD conformation can be acquired by rotating the ECD roughly 30° horizontally from its position in the crystal structure (Fig. 5d). These simulations indicate that the ECD in the apo-state GLP-1R is quite dynamic and that the TMD-interacting closed conformations are energetically favorable in the absence of the peptide agonist. Therefore, we conclude our crystal structure represents a physiologically relevant snapshot of GLP-1R in an inactive peptide-free state.

**Comparison of inactive GLP-1R and GCGR.** Comparison of the inactive GLP-1R–Fab7F38 structure with the previous inactive GCGR–mAb1 structure[14] reveals a remarkable similarity of the inner halves of the TMD, which is consistent with the high sequence identity and common Gs coupling of the two receptors. Likewise, the helix IV–ECL2–helix V region displays a high degree of structural similarity between the two inactive receptors. In contrast, major structural differences are observed in the stalk, ECL1, ECL3, and ECD (Fig. 6a, b). The Fab7F38-bound GLP-1R is crystallized in a closed conformation with the ECD's peptide-binding area sealed, in line with the non-competitive property of Fab7F38. The ECD of GCGR, on the other hand, is in an extended

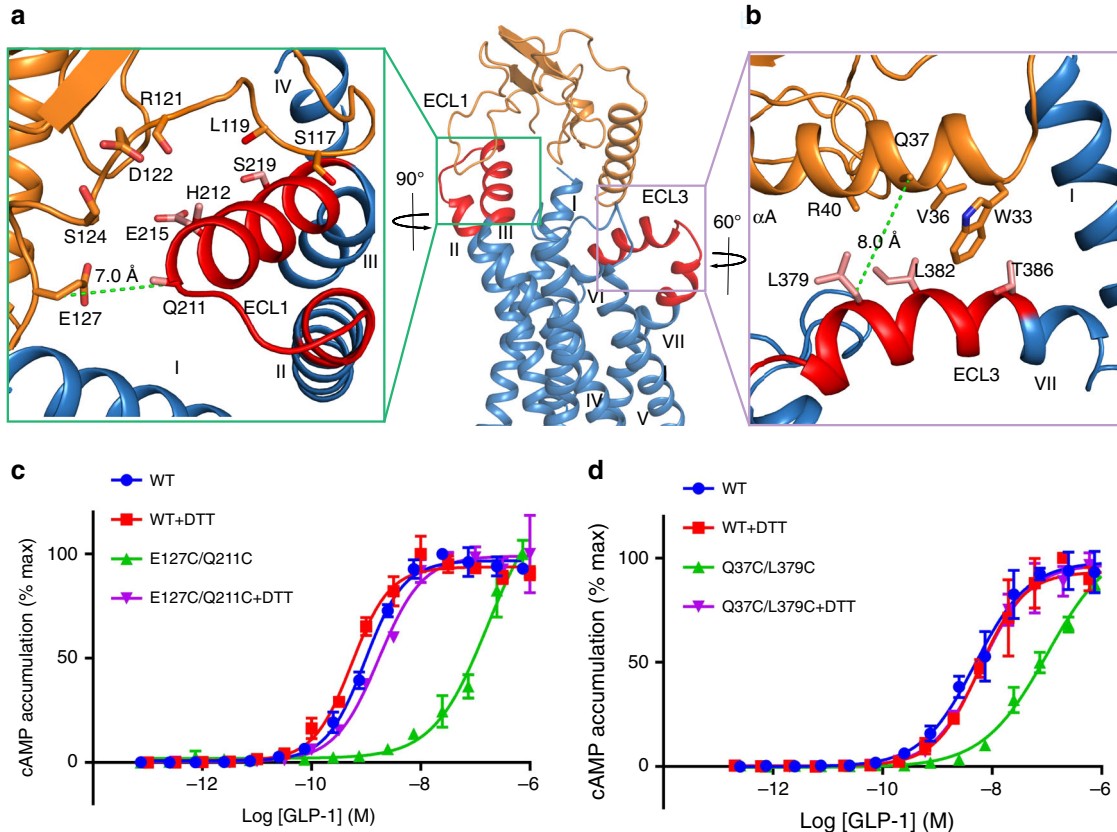

**Fig. 3 ECD-ECL1/3 disulfide crosslinking studies. a, b** Interactions between the ECD and ECL1 or ECL3 in the GLP-1R–Fab7F38 complex structure. ECD, ECL1/3, and TMD are colored orange, red and blue, respectively. The interaction residues are shown as sticks. The Cβ-Cβ distances of Q37-L379 and E127-Q211 are given and marked with green dashed lines. **c, d** Disulfide crosslinking studies of the GLP-1R double mutant E127C/Q211C and Q37C/L379C. GLP-1–induced cAMP measurement of mutants E127C/Q211C (**c**) and Q37C/L379C (**d**) with or without the presence of 1 mM DTT. Dose-response curves of cAMP accumulation assays were generated and graphed as mean ± s.e.m. from three independent experiments each performed in duplicate. Wild-type samples were used as control. Source data are provided as a Source Data file.

conformation with the β-sheet module covering the orthosteric TMD pocket and the inhibitory mAb1 engaging the orthosteric ECD-binding site directly. Clearly, mAb1-bound GCGR is unable to assume the closed conformation observed in the Fab7F38–GLP-1R structure. The closed conformation of the inactive GLP-1R results in partial solvent protection of the hydrophobic ECD binding site, which is considered energetically favorable for an apo-state compared to an open conformation with full solvent exposure of the binding site. In the inactive GCGR structure, ECL1 forms a β-hairpin conformation and runs in parallel with the stalk to form a compact β-sheet module. Transition to the glucagon-bound state requires the lid-like β-sheet module to undergo a major conformational change. Both the stalk and ECL1 transform to a short α-helix and form extensive interactions with the peptide, as shown in the previously reported peptide-bound active GCGR structure[9]. In comparison, in GLP-1R the ECL1 consistently adopts a α-helical conformation and the ECL1, ECL3 and stalk of the TMD all reorient to facilitate the binding of the peptide agonist. Importantly, the ECD of GLP-1R undergoes a remarkable orientation change from the closed TMD-interacting conformation to the extended active conformation during peptide binding and activation. The observed interactions between the ECD and ECL1/3 in the inactive GLP-1R structure presented here agree with previous studies suggesting that the ECD stabilized an inactive state of the glucagon receptor through interactions with ECL1 and ECL3[20].

Different types of antibodies facilitated crystallization of GLP-1R and GCGR into varied ECD orientations, whereas the

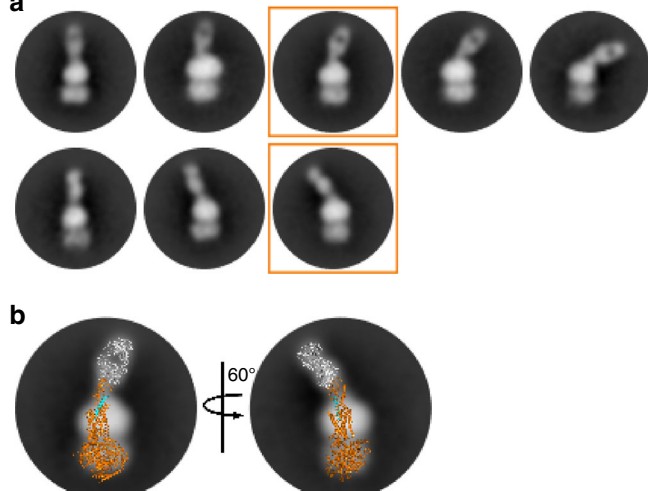

**Fig. 4 Single-particle EM analysis reveals multiple conformations of the Fab7F38–semaglutide–GLP-1R–Gs–Nb35 complex. a** 2D class averages of Fab7F38–semaglutide–GLP-1R–Gs–Nb35 complex in negative stain EM showing conformational flexibility of the Fab7F38-bound ECD. It cannot be excluded that the observed motions of the Fab-bound ECD are bilateral. **b** A model of the Fab7F38-bound active state GLP-1–GLP-1R–Gs complex is superimposed on to densities of the 2D class averages highlighted in a. Fab7F38 is shown in grey, GLP-1R–Gs in orange, and GLP-1 in cyan. The rotation arrow refers to the active model being rotated to fit the two 2D classes.

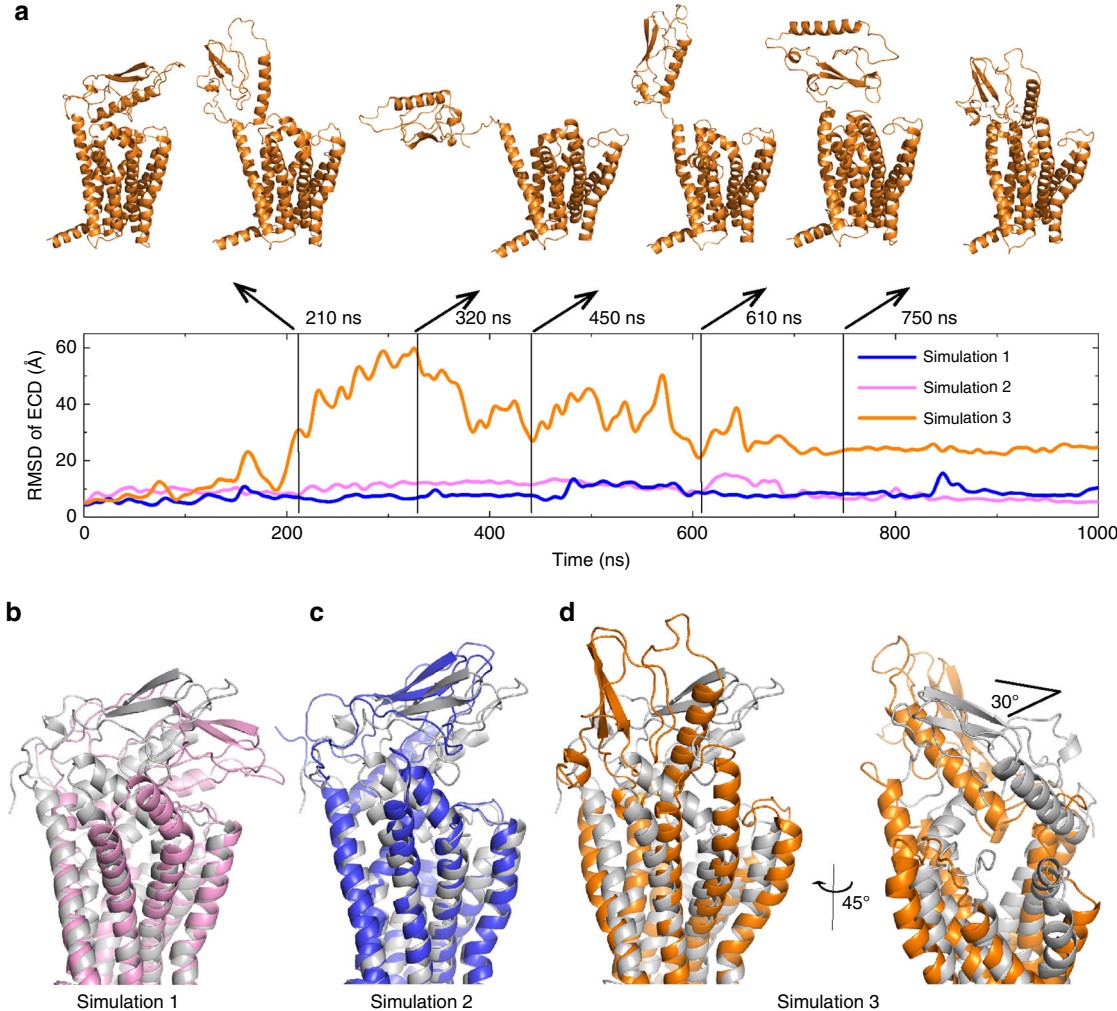

**Fig. 5 MD simulations of apo GLP-1R. a** Main chain r.m.s.d values of the ECD versus simulation time in the three 1-μs MD simulations and typical snapshots of trajectory 3. The values were calculated from snapshots at 100 ps intervals. All structures are superimposed on the crystal structure of full-length GLP-1R using the main chain atoms of residues A153[1.48b]–S163[1.58b] (helix I) (residues are designated in superscript according to the Wootten residue numbering scheme[49]), I179[2.49b]–I196[2.66b] (helix II), V229[3.32b]–G248[3.51b] (helix III), G273[4.49b]–P277[4.53b] (helix IV), T353[6.42b]–I357[6.46b] (helix VI), and Q394[7.45b]–Y402[7.53b] (helix VII). **b–d** Comparisons between simulation snapshots and full-length GLP-1R crystal structure (grey cartoon). Each snapshot represents an average conformation of the last 300 ns of each trajectory.

structural differences in the TMD extracellular regions, including the stalk of helix I, is potentially due to sequence diversity between the two receptors. While ECL2 and ECL3 are relatively conserved within class B receptors[21], the stalk and ECL1 regions are highly variable between GLP-1R and GCGR (Fig. 6c). The GLP-1R stalk sequence [126-]EESKRGERSS[−137] is highly entropic, which may contribute to its disorder in current and previous GLP-1R–Gs structures. In contrast, the GCGR stalk residues are amphipathic and were resolved in all full-length GCGR structures. Furthermore, in ECL1 of GCGR there are two glycine residues (G207[ECL1], G219[ECL1]) that disconnect the helical structure in the active state, whereas the G219[ECL1] is not conserved in GLP-1R and the helical conformation is relatively intact in this region of GLP-1R. Importantly, both the stalk and ECL1 of GCGR contain several β-branched residues that favor the β over α conformation, in contrast to only one β-branched residue (T207) in ECL1 and none in the stalk region in GLP-1R. These distinct sequence features may provide the basis for the different structural modules of GLP-1R and GCGR, as well as diverse pathways for conformational change during peptide binding and activation.

## Discussion

The crystallized inactive GLP-1R structure is not compatible with the known binding mode of GLP-1 requiring a conformational change of the ECD and a rearrangement of the TMD to enable binding of GLP-1. The observed closed conformation of GLP-1R suggests a feasible mechanism for its conversion to an extended open conformation for peptide recognition. We propose that in the absence of a peptide, the ECD is dynamic but favors a closed conformation stabilized by the weak interactions between the ECD and ECL1/3. Subtle conformational dynamics allow the C-terminus of GLP-1 to access the ECD binding site, and the ligand binding triggers further dissociation of the ECD from the TMD allowing the N-terminus of GLP-1 to enter the orthosteric pocket in the TMD and activate the receptor. Alternatively, a pre-existing small population of open conformations can accommodate the peptide hormone smoothly and this may trigger the transition of GLP-1R from the closed to the otherwise energetically unfavorable open conformation (Fig. 7). This scenario was illustrated in our previous dynamic study on GCGR[15]. In support of this hypothesis, locking the ECD–TMD interactions with disulfide bonds compromises the functional efficiency of GLP-1.

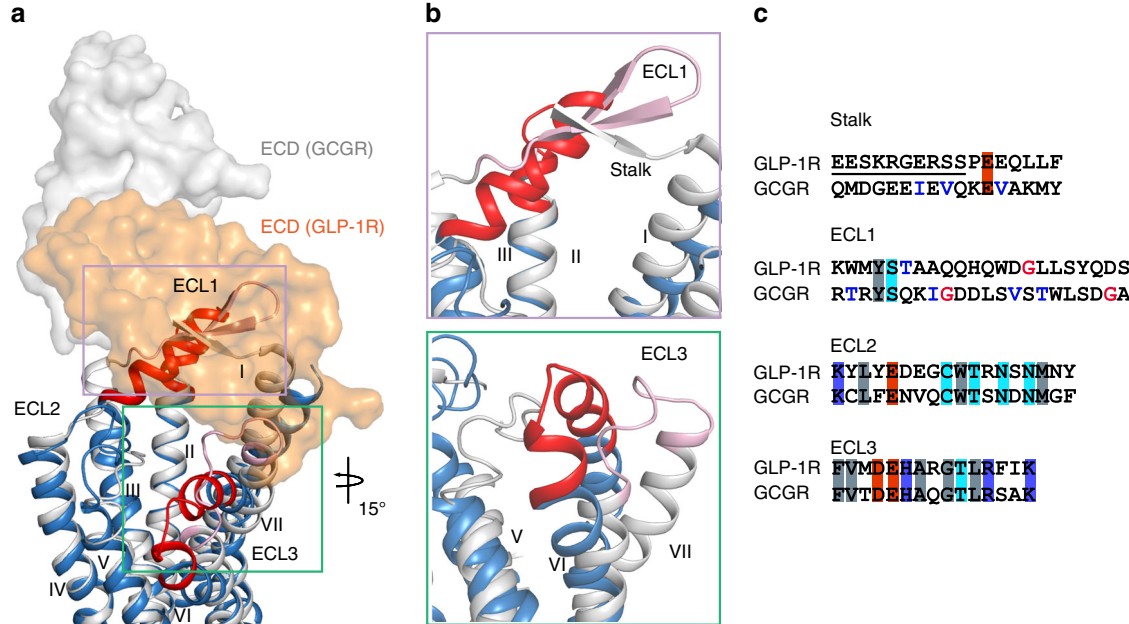

**Fig. 6 Comparison of inactive GLP-1R and GCGR. a** Different structural motifs of the stalk, ECL1 and ECL3 between GLP-1R and GCGR in their respective inactive conformations. TMDs of GLP-1R and GCGR are shown as blue and grey cartoons, respectively, while their ECL1 and ECL3 are shown as red (GLP-1R) and pink (GCGR) cartoons, and ECDs are labelled and shown as surfaces. **b** Zoom-in views of the key differences in stalk, ECL1 and ECL3. **c** Alignments of key residues in stalk, ECL1, ECL2 and ECL3 between GLP-1R and GCGR. β-branched and glycine residues are colored blue and red, respectively. The conserved hydrophobic, neutral hydrophilic, acidic and basic residues are shown with grey, cyan, red and blue backgrounds, respectively. Underline indicates the highly entropic region in GLP-1R.

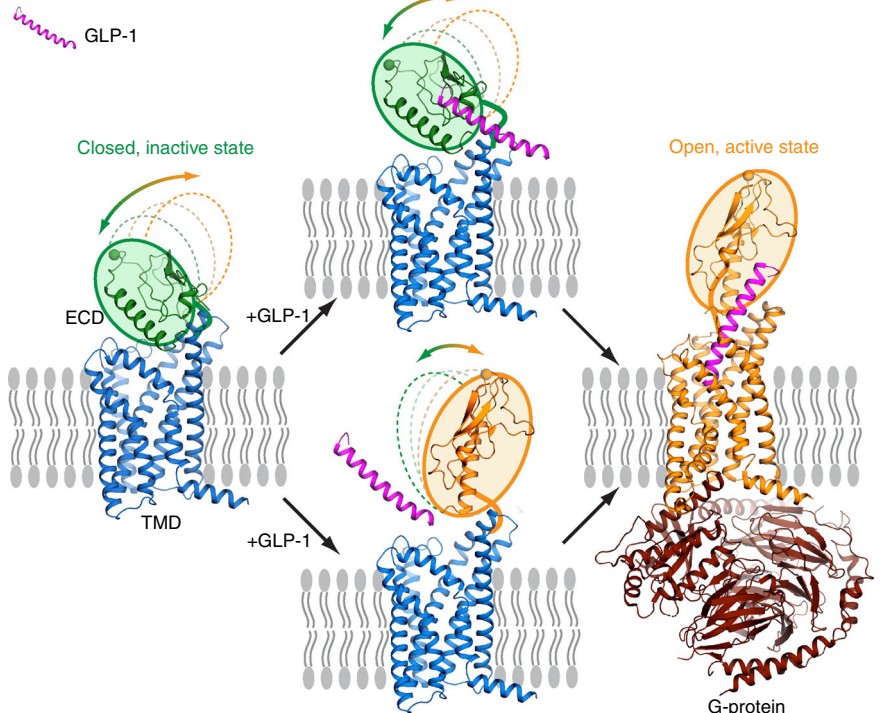

**Fig. 7 The canonical two-domain activation pathway.** Without the presence of GLP-1 (upper left), the receptor is dynamic and the ECD can adopt multiple conformations (dashed circles) but favors a closed inactive state (solid green circle). The subtle dynamics of the ECD allows binding of GLP-1's C-terminus to the ECD (upper), which triggers further dissociation of the ECD from the TMD allowing GLP-1's N-terminus to enter the orthosteric pocket in the TMD and activate the receptor (right). Alternatively, the pre-existing small population of open conformations (solid yellow circle) can accommodate GLP-1 smoothly (lower) and then trigger the transition of GLP-1R from the closed to open conformation to accommodate the downstream G-protein (right). The ECD movements are indicated with arrows, and the closed and open conformations are colored with green and yellow backgrounds, respectively. The inactive (current structure) and active (PDB: 5VAI) TMDs are shown as blue and yellow cartoons, respectively; G protein from the active structure (PDB: 5VAI) is shown as brown cartoon. Cell membranes are shown as grey lipid bilayers.

The ECD has been proposed to function as a negative regulator of GCGR[20]. Specifically, mutagenesis of ECD, ECL1 and ECL3 individually increased the basal activity of GCGR, which could result from the interruption of intramolecular interactions between the ECD and ECL1/3 that otherwise stabilize the receptor in an inactive state. The inactive structure of GLP-1R presented here aligns nicely with a possible role of the GLP-1R ECD as a negative regulator of GLP-1R, although, in contrast to GCGR, we and others have failed to identify mutations in the ECD/ECL interface that increase basal activity of GLP-1R. However, this could reflect differences in the TMD of GLP-1R and GCGR and their intrinsic ability to activate G protein in the absence of agonist, independent of the ECD. Obviously, the ECD of the closed conformation prevents GLP-1 binding and further experiments are necessary to fully understand the functional impact of the closed ECD conformation on Gs binding. The multiple conformations observed from the single-particle EM analysis of the purified Fab7F38–semaglutide–GLP-1R–Gs–Nb35 complex most likely represent different agonist-bound ECD conformations, however some 2D classes are also compatible with the closed conformation observed in our inactive crystal structure (Supplementary Fig. 5d). We speculate that a fraction of the purified receptor complex may bind Gs in the absence of the agonist. Although less likely, such a scenario is not unusual, and has been visualized in the prototypical β2 adrenergic receptor: both ligand-free and antagonist-bound β2 adrenergic receptors can form a complex with nucleotide-free Gs[22,23]. The non-competitive Fab7F38 could be an important tool for further structural clarification of the existence of different states of GLP-1R.

According to the model presented here based on current inactive and active structures of GLP-1R, the canonical peptide activation pathway involves a major conformational change of the ECD, going from a closed inactive to open active state (Fig. 7). Non-canonical activation mechanisms may exist and the ECD could play a different role assuming different conformations depending on the ligand[24]. Most recently, in a new structure of GLP-1R, a small molecule agonist, TT-OAD2, was shown to activate the receptor through binding of the helix II–ECL1–helix III region, revealing another ECD conformation[25]. The increased understanding of activation pathways and related ECD conformations may inspire the design of new molecules targeting the ECD–ECL interface.

In summary, the reported inactive GLP-1R crystal structure reveals a unique closed conformation of the ECD, and a deeper understanding of the conformational dynamics of GLP-1R was elaborated by EM and MD simulations. The inactive structure aligns with the dual functional hypothesis of the ECD through which, on one hand, peptide binding and activation is mediated when the ligand is present, while on the other, the receptor is negatively regulated in its peptide-free state. The structure enables a comparison of the only two inactive peptide-free class B GPCR structures and increases our structural understanding of their activation pathways, which may be helpful in the design of novel ligands, thereby enabling new avenues for drug discovery on therapeutically important class B GPCRs.

## Methods

**Construct modification and expression of full-length GLP-1R.** The optimized gene of full-length GLP-1R was cloned into the modified pFastbac1 vector at the BamHI/HindIII sites and the native signal sequence was replaced by haemagglutinin (HA) to enhance receptor expression (see primers in Supplementary Table 2), followed by a flag-tag, a 10× His-tag and a tobacco etch virus protease (TEV) site at the N-terminus. To increase the stability of GLP-1R, the fusion protein rubredoxin was inserted into intracellular loop 2 (ICL2) of GLP-1R between positions 257 to 262 and 11 mutations were introduced. Besides the 10 thermostabilizing mutations that reported previously[26,27], E387[7.42b]D was introduced to mimic the identified interaction in the GCGR-NNC1702 complex structure[9]. The C-terminus was truncated to 437 to further increase the thermostability of receptor. The construct was expressed in *Spodoptera frugiperda* (*Sf*9) insect cells (Invitrogen) with Bac-to-Bac Baculovirus system (Invitrogen) and cells were infected at a density of $2 \times 10^6$ cells per mL and

collected 48 h after infection. Our initial attempts to crystallize the engineered full-length GLP-1R in complex with PF-06372222 (a previously used negative allosteric modulator for crystallization of the isolated TMD) did not yield any crystal. However, using an ECD-binding antibody Fab fragment (Fab7F38) for co-crystallization, we successfully crystallized the GLP-1R–PF-06372222–Fab7F38 complex.

**Expression and purification of Fab7F38.** ExpiCHO-S™ cells (mouse, Chinese hamster ovary) were cultured in serum free ExpiCHO™ Expression Medium (Gibco). Transient transfection of pJSV-based vectors with heavy chain and light chain of Fab7F38 was carried out using ExpiFectamine™ CHO Reagent (Gibco). The culture supernatant was harvested by centrifugation at 6000 *g* for 20 min and clarified by filtration. Fab7F38 was affinity-captured by a Protein G Sepharose 4FF column (GE healthcare) and eluted with a low pH elution buffer (100 mM Glycine pH 2.8). The eluted sample was quickly neutralized by addition of 1/10 volume of 1 M Tris pH 8.0 and further polished on a size-exclusion chromatography column (Superdex 75, GE healthcare) pre-equilibrated with phosphate-buffered saline (PBS, pH 7.4). The main peak eluted from the SEC column correlated with the target Fab7F38 protein, was pooled and stored in −80 °C. Protein concentration was determined by A280 measurement.

**Purification of GLP-1R–PF-06372222–Fab7F38 complex.** The 1 L cell biomass expressing modified GLP-1R construct was lysed in a low salt buffer containing 10 mM HEPES pH 7.5, 20 mM KCl, 10 mM MgCl₂, and EDTA-free protease inhibitor cocktail tablets. The sample was then centrifuged at 160,000 *g* for 35 min to collect the membranes. The membranes were washed three times in a high salt buffer containing 10 mM HEPES pH 7.5, 1 M NaCl, 20 mM KCl, and 10 mM MgCl₂. Purified membranes were resuspended in 40 mL low salt buffer and incubated with 100 μM PF-06372222, 2 mg mL⁻¹ iodoacetamide, and EDTA-free protease inhibitor cocktail tablet for 1 h at 4 °C. The protein sample was extracted from membrane by adding a 2× solubilization buffer containing 20 mM HEPES pH 7.5, 500 mM NaCl, 2% (w/v) n-dodecyl-beta-D-maltopyranoside (DDM, Affymetrix), 0.4% (w/v) cholesteryl hemisuccinate (CHS, Sigma), and 2% (w/v) glycerol for 3 h at 4 °C. The sample was centrifuged at 160,000 *g* for 35 min and the supernatant was incubated with 1 mL TALON resin (Clontech) and 20 mM imidazole overnight at 4 °C. The resin was washed by 20 column volumes of wash buffer A [20 mM HEPES pH 7.5, 500 mM NaCl, 2% (w/v) glycerol, 0.05% (w/v) DDM, 0.01% (w/v) CHS and 30 mM imidazole] and 10 column volumes of wash buffer B [20 mM HEPES, pH 7.5, 500 mM NaCl, 2% (w/v) glycerol, 0.02% (w/v) DDM, 0.01% (w/v) CHS and 50 mM imidazole], followed by incubation with Fab7F38 at a molar ratio of 1: 1.5 in 3 mL wash buffer C [20 mM HEPES pH 7.5, 500 mM NaCl, 2% (w/v) glycerol, 0.01% (w/v) DDM, 0.01% (w/v) CHS and 20 mM imidazole] for 3 h at 4 °C. The unbound Fab7F38 was removed by 5 mL wash buffer C. The resin was resuspended by 2 mL wash buffer C and the TEV protease was added to remove the N-terminal tag at a molar ratio of 1:10 and the mixture was shaken at 4 °C for at least 16 h. The GLP-1R–PF-06372222–Fab7F38 complex was collected from the flow-through of the resin and concentrated to ~40 mg mL⁻¹ for crystallization trials.

The protein sample was mixed with lipid (monoolein/cholesterol 10:1 by mass) at weight ratio of 2:3 using a syringe mixer. The lipidic cubic phase (LCP) mixture was dispensed onto 96-well glass sandwich plates in 50 nL drops and overlaid with 800 nL precipitant solution using a NT8 (Formulatrix). The crystals appeared in 200–300 mM ammonium formate, 36% PEG400, 5–10% (w/v) guanidine hydrochloride, pH 6.2–6.6 after 7 days and reached their biggest size (~150 μm) in 1 month. Crystals were harvested directly from LCP using 50–150 μm micromounts (M2-L19-50/150, MiTeGen), flash frozen, and stored in liquid nitrogen.

**Data collection and structure determination.** X-ray diffraction data were collected at the Spring8 beam line 45XU, Hyogo, Japan, using a Rayonix 10 × 10 μm minibeam for 0.1 s and 0.1°–0.5° oscillation per frame. Data of most crystals were limited to 10° because of radiation damage and only two crystals were collected to 180°. The collected images were automatically processed with KAMO[28], and XDS[29] was used for integrating and scaling data from the 21 best-diffracting crystals for GLP-1R–PF-06372222–Fab7F38. The GLP-1R–PF-06372222–Fab7F38 complex was solved by molecular replacement with Phaser[30] using the models of active-like GLP-1R crystal structure (PDB: 5NX2) and rubredoxin (PDB: 1FHH). The structure was refined using Phenix[31] and Buster[32] with manual examinations of 2Fo-Fc and Fo-Fc maps with Coot[33]. The final model of GLP-1R–PF-06372222–Fab7F38 contained 29–257, 315–474 of GLP-1R, 258–314 residues of rubredoxin, and chain B and chain C of Fab7F38. All three ECLs and ICLs were well-resolved, whereas the stalk was disordered. The buried areas of domain −domain or peptide−receptor interfaces were calculated with PDBePISA[34]. The pocket volumes of inactive (GLP-1R–Fab7F38), active-like (GLP-1R–peptide 5), and active (GLP-1R–GLP-1–Gs) structures were calculated with POVME[35].

**cAMP accumulation assay.** Wild-type GLP-1R and GLP-1R mutants were cloned into the expression receptor pcDNA3.1/V5-His-TOPO vector (Invitrogen) at the HindIII and EcoRI sites by using the QuickChange site-directed mutagenesis and a flag-tag was inserted after native signal sequence (see primers in Supplementary Table 2). Sequences of receptor clones were confirmed by DNA sequencing. This GLP-1R construct had equivalent pharmacology to the untagged human GLP-1R

based on cAMP assays. HEK293T cells (obtained from and certified by the Cell Bank at the Chinese Academy of Sciences and confirmed as negative for mycoplasma contamination) were cultured in DMEM supplemented with 10% (v/v) foetal bovine serum, 50 IU mL$^{-1}$ penicillin and 50 µg mL$^{-1}$ streptomycin. Cells were maintained in an incubator at 37 °C and 5% CO$_2$ incubator and seeded onto 6-well cell culture plates before transfection. After overnight culture, the cells were co-transfected with pGlo-Sensor™−22F cAMP plasmid (Promega) and GLP-1R DNA using Lipofectamine 2000 transfection reagent (Invitrogen). After 24 h, expression levels of mutants were measured to be 90%–100% of the wild-type for follow-up experiments. The transfected cells were seeded onto 384-well plates (15,000 cells per well). After overnight culture, transfected cells were incubated with 200 µg mL$^{-1}$ D-luciferin (BioVision) in an incubator at 37 °C and 5% CO$_2$ for 1 h. The cells were treated with Hank's Balanced Salt Solution (Hyclone) or 1 mM DTT for 10 min before incubation in Hank's Balanced Salt Solution with different concentrations of GLP-1 (0–2 µM) at room temperature for 20 min. Chemiluminescence signals were measured at 620 and 650 nm by an EnVision plate reader (PerkinElmer).

The lack of a Fab7F38 effect on GLP-1 mediated cAMP production was tested in H293T stably overexpressing SNAP-tagged human GLP-1R (the plasmid purchased from Cisbio). The cells were seeded onto 384-well plates (2000 cells per well). After 48 h, the cells were washed with PBS (Gibco) and incubated in the assay buffer (HBSS supplemented with Ca$^{2+}$ and Mg$^{2+}$ (Gibco), 10 mM HEPES (Gibco), 0.1% Pluronic F-68 (Gibco) and 500 µM IBMX (Sigma), pH 7.4) with different concentrations of GLP-1 (0 µM to 1 µM) in the presence (100 nM and 1000 nM) or absence of Fab7F38 at 37 °C for 30 min. The cAMP accumulation was measured using a cAMP Gs Dynamic Kit (Cisbio) and EnVision reader (PerkinElmer).

**Expression and purification of Fab7F38–GLP-1R–semaglutide–Gs.** Human GLP-1R (Arg24-Leu422) fragment fused with a N-terminal HA-FLAG-B2-HRV14-3C tags and C-terminal HPC4-HRV14-3C-10× His tags was synthesized and inserted into a modified pFastBac-1 vector (BamHI/HindIII sites) under the polh promoter (Genewiz) (see primers in Supplementary Table 2). A dominant-negative Gαs (DNGαs) was generated[3] by introducing eight mutations: S54N, G226A, E268A, N271K, K274D, R280K, T284D, and I285T. Human Gβ1 and Gγ2 were inserted into a pFastDual plasmid and were prepared by insertion of His$_6$-tagged human Gβ$_1$ fragment (BamHI/HindIII site) under the polh promoter and insertion of Gγ$_2$ fragment (XhoI/SphI site) under the p10 promoter, respectively (Genewiz).

Human GLP-1R, human DNGαs, and His$_6$-tagged human Gβ$_1$ and Gγ$_2$ were expressed in sf9 insect cells (Expression systems) using baculovirus and then infected with three separate baculoviruses at a ratio of 5:2:2 for GLP-1R, DNGαs and Gβ$_1$γ$_2$. The cells were infected at a density of 2 × 10$^6$ cells mL$^{-1}$ and collected 48 h after infection and cell pellets were stored at −80 °C. The cell pellet was thawed, homogenized, and lysed in a buffer containing 20 mM HEPES pH 7.5, 50 mM NaCl, 2 mM MgCl$_2$, and complete protease inhibitor cocktail tablet (Roche), and the membranes were collected by centrifugation at 40,000 g. This step was repeated one additional time before the GLP-1R–semaglutide–Gs complex was formed by the addition of 10 µM semaglutide, 10 µg mL$^{-1}$ Nb35, and 25 mU mL$^{-1}$ apyrase to the homogenized membrane suspension. The suspension was incubated for 1 h at room temperature, before the membranes were isolated by centrifugation at 40,000 g for 30 min. The membrane-bound complex was solubilized by 0.5% (w/v) lauryl maltose neopentyl glycol (LMNG) supplemented with 0.03% (w/v) cholesteryl hemisuccinate (CHS) for 2 h at 4 °C in the presence of 10 µM semaglutide, 10 µg mL$^{-1}$ Nb35, and 25 mU mL$^{-1}$ apyrase. Insoluble material was removed by centrifugation at 40,000 g for 30 min, and the supernatant was incubated with HPC4 resin and 5 mM CaCl$_2$ overnight at 4 °C. The receptor-bound HPC4 resins were washed with a buffer containing 20 mM HEPES pH 7.5, 100 mM NaCl, 2 mM MgCl$_2$, 0.01% (w/v) LMNG, 0.006% (w/v) CHS, and 5 µM semaglutide supplemented with 5 mM CaCl$_2$ before the bound material was eluted in a buffer supplemented with 10 mM EDTA. The eluted protein was concentrated using an Amicon Ultra Centrifugal Filter (MWCO 100 kDa), and Fab7F38 was added at an estimated 1:1 molecular ratio of Fab–receptor complex. The complex was subjected to size-exclusion chromatography (SEC) on a Superose 6 Increase 10/300 column (GE Healthcare) equilibrated with 20 mM HEPES pH 7.5, 100 mM NaCl, 2 mM MgCl$_2$, 1 µM semaglutide, 0.01% (w/v) LMNG, and 0.006% (w/v) CHS. The eluted peak fraction containing the Fab7F38–GLP-1R–semaglutide–Gs was diluted to 50 µg mL$^{-1}$ and used for single-particle negative stain EM analysis.

**Negative stain electron microscopy.** A 3 µL sample was placed onto a carbon-coated, glow-discharged, 300-mesh copper grid for 30 s, followed by staining three times with 3 µL 2% uranyl formate solution. Images were acquired with a FEI Tecnai Spirit Twin (120 kV) equipped with a Tietz F416 camera using Leginon[36]. 444 images were collected with a nominal underfocus of 0.7–1.7 µm at ×67,000 magnification with a binned camera (pixel size 3.15 Å).

The acquired micrographs were inverted using the EMAN2[37] to facilitate particle picking in cryoSPARC[38]. The remaining data processing was performed using the cryoSPARC v2 workflow. In brief, CTF was estimated using CTFFIND4[39] after which 427 micrographs were accepted by manual inspection. Particle picking was conducted first by manual picking of 400 particles, which were extracted using box size 104 and subjected to 2D classification. The obtained "good" 2D classes were used for template-based automatic particle picking, and false positives or "bad particles" were subsequently eliminated over two rounds of 2D classification,

resulting in 9101 selected particles. The model of the open state conformation was based on the cryo-EM structure of GLP-1–GLP-1R–Gs (PDB: 5VAI) and the Fab7F38–ECD complex of the Fab7F38–GLP-1R crystal structure presented here, which were superimposed on their ECDs using PyMOL.

**MD simulations.** Initial wild-type GLP-1R model for MD simulations was obtained from the GLP-1R-Fab7F38 structure with modifications: (1) antibody (Fab7F38) and fusion protein (rubredoxin) were removed; (2) 11 thermostabilized mutations were mutated back to wild-type residues; 3) missing residues in N-terminus (R24-A28), stalk (S129-R134), and ICL2 (S258-S261) were modelled using MODELLER[40]. The N-terminus was positively charged, and the C-terminus was capped with neutral groups. This GLP-1R model was embedded in a 95 × 95 Å palmitoyl oleoyl phosphatidyl choline (POPC) bilayer and lipids located within 1 Å of the receptor were removed. The system was solvated in a box (95 × 95 × 146 Å) with TIP3P waters and 0.15 M NaCl, including 241 lipid molecules, 26064 water molecules, 77 chloride ions, and 73 sodium ions, for a total of 117,175 atoms.

MD simulations were performed using the GROMACS 2018 package[41] with isothermal–isobaric (NPT) ensemble and periodic boundary condition. The CHARMM36 force field[42] was used for the protein, glucagon, the POPC phospholipids, ions and water molecules. Energy minimizations were performed to relieve unfavourable contacts in the system, followed by equilibration steps of 50 ns in total to equilibrate the lipid bilayer and the solvent with restraints on the main chain or C$_\alpha$ atoms of GLP-1R. Subsequently, three 1-µs production runs were performed. The temperature of the system was maintained at 310 K using the Nose–Hoover method[43,44] with a coupling time of 0.5 ps. The pressure was kept at 1 bar using the Parrinello–Rahman[45] with τ$_p$ = 2.0 ps and a compressibility of 4.5×10$^{-5}$ bar$^{-1}$. SETTLE[46] and LINCS[47] constraints were applied on the hydrogen-involved covalent bonds in water and other molecules, respectively, and the time step was set to 2 fs. Electrostatic interactions were calculated with the Particle-Mesh Ewald (PME) algorithm[48] with a real-space cut-off of 1.2 nm.

**Reporting summary.** Further information on research design is available in the Nature Research Reporting Summary linked to this article.

## Data availability

Data supporting the findings of this manuscript are available from the corresponding authors upon reasonable request. A reporting summary for this Article is available as a Supplementary Information file. The source data underlying Figs. 1c, 3c, d and Supplementary Figs. 4a–d, 5a are provided as a Source Data file. Atomic coordinates and structure factors for the GLP-1R–Fab7F38 structure has been deposited in the Protein Data Bank with identification code 6LN2.

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

## Acknowledgements

The authors thank the Cloning, Cell Expression, Protein Purification and Assay Development Core Facilities of iHuman Institute for their support. We thank the Shanghai Municipal Government, ShanghaiTech University and GPCR Consortium for financial support. We thank Qingtong Zhou and Yuxia Wang for their assistence in model analysis or protein preparation, and Angela Walker for assistance with manuscript preparation. This work was also supported by the National Key Research and Development Program of China (2018YFA0507001, 2018YFA0507000, 2018YFA0508100), the National Natural Science Foundation of China grants 31770898 (G.S.), 31900895 (L.Y.) and 81872915 (M.-W.W.), the National Science and Technology Major Project "Key New Drug Creation and Manufacturing Program" of China (2018ZX09735-001) and China Postdoctoral Science Foundation (2017M622365). The synchrotron radiation experiments were performed at the BL45XU of Spring-8, Japan. Computational resources were provided by Henan Supercomputer Center. The EM data were collected at iNANO Cryo-Electron Microscopy Facility (EMBION), Aarhus University, Denmark and with the support of facility manager Dr. Thomas Boesen. R.C.S. acknowledges that U.S.C. is his primary affiliation.

## Author contributions

F.W. optimized constructs, expressed and purified proteins, crystallized and determined the structure, and wrote the initial manuscript; L.Y. performed and analyzed MD simulations on wild-type and mutant GLP-1Rs. K.H. performed mutagenesis and functional assays. M.L. collected and processed negative stain EM images. L.W. and G.W.H. helped on crystal data collection and structure refinement. Q.R. did the insect cell expression of GLP-1R/Gs and Nb35 for EM study on GLP-1R. N.K.R. helped on functional experiments with GLP-1R and Fab7F38. G.L. assisted receptor purification at early stage of this project. M.A.H. provided advice on crystallization and edited the manuscript. H.J. oversaw the MD simulations. M.-W.W. helped with compound synthesis, oversaw the functional assays and edited the manuscript. S.R.-R. provided Fab7F38 and advice on crystallization trials and edited the manuscript. G.S. helped mentor the project and edited the manuscript. R.C.S. conceived the overall project during an initial sabbatical and guided it through to completion. All authors were involved in the discussions and provided comments on the manuscript.

## Competing interests

S.R.-R. is an employee of Novo Nordisk, a pharmaceutical company focused on GLP-1R for type 2 diabetes. R.C.S. is a founder and board member of Bird Rock Bio, a company focused on GPCR therapeutic antibodies. R.C.S. is a founder, board member, and employee of ShouTi, a company focused on GPCR small molecules. M.A.H. is a founder and employee of ShouTi. The remaining authors declare no competing interests.
