## [Peer Review File · Nature Communications]

Reviewers' Comments:

Reviewer #1:

Remarks to the Author:

This is a very interesting paper revealing new features regarding the mechanism of action of Family B GPCRs.

One minor point to address.

While I understand what the authors are trying to convey, the statements on p10 referring to the ECD being an "antagonist" do not make pharmacological sense.

"..the receptor is antagonized in its apo-state. The antagonism feature of the ECD revealed here indicates..."

An antagonist is defined as a ligand that reduces the action of another ligand (an agonist), which is clearly not the case in the situation with the apo-state GLP-1R. The language needs to be modified to avoid misusing this pharmacological term.

Reviewer #2:

Remarks to the Author:

This paper reports a novel crystal structure of the GLP-1R in a peptide free state that likely represents the inactive conformation of the receptor. It reveals a unique conformation of the ECD that is closed over the TM bundle. This structure is also supported by crosslinking data confirming specific interactions within the ECD and ECLs that stabilise the closed conformation. They also use a series of MD experiments to propose a model of class B GPCR activation by peptides and use negative stain EM, MD and previously published structures to provide a model for class B GPCR activation by peptide agonists. While this paper itself does not change our current understanding of class B GPCR activation by peptide agonists, it does solidify our understanding. A closed conformation of the receptor ECD in the apo state has been predicted for some time and is consistent with earlier MD studies on the GCGR and GLP-1R and mutagenesis studies on the SecR and GCGR that supported the notion that the inactive state is stabilised by a closed ECD conformation mediated by ECD-ECL3 interactions that need to be released for peptide engagement. This is the first time that this closed conformation of the receptor has been experimentally validated and visualised, revealing not only extensive interactions between the ECD and ECL3, but also with ECL1 and therefore represents a significant advance for the class B GPCR field. I have a number of comments on the manuscript in its current form.

Main comments

In the title and abstract, this structure is referred to as an apo-GLP-1R. While the authors speculate that the structure represents an apo state, the structure is bound to both a NAM and an antibody and is therefore by definition not an apo structure. In addition, the receptor has been stabilised in an inactive conformation (by both the use of a NAM and mutagenesis) and therefore is unlikely to sample the conformational landscapes occupied by the apo receptor. This should therefore be revised to state either inactive or peptide-free, not apo.

One caveat to this study is that the receptor contains 11 thermostabilising mutations including a disulphide stabilising TM5-6 and mutations at the top of TM7 that may influence the conformation, particularly the location of ECL3, however given the authors have shown convincing data supporting the interactions in the x ray structure, this is not too much of a concern. Did the authors try generating crystals with a construct containing less mutations?

In this structure PF06372222 was used as a NAM to stabilise the structure. It should be clearly stated (currently it is not mentioned anywhere) that this is not a NAM of the native GLP-1R (it is a GCGR NAM) and only behaves as a NAM in the modified construct used in this study that contains residues from the GCGR that allow this to bind the GLP-1R. The authors have used this construct and NAM previously, and there is some confusion in the literature where this compound has been reported as a GLP-1R NAM, where it actually isn't. If the authors are using a tool compound that only behaves at a modified non-native receptor, it is their duty to make this clear in the paper.

In figure 4 (and the related text) the authors show low resolution negative stain EM data of semaglutide bound the GLP-1R:Gs complex. Here they speculate that they have two different states, one bound by semaglutide and one unliganded and they have therefore docked their inactive model into one of these classes even though the class average clearly contains G protein). They speculate that the GLP-1R can bind Gs in the absence of ligand. This section is highly speculative and is not supported by any experimental data. It is not appropriate to state the GLP-1R can adopt an intracellular active conformation capable of coupling to G protein without agonist peptide binding, when there is absolutely no data presented to support this. While it is possible, it is unlikely to be the case here. The complex has been stabilised using a high concentration of semaglutide, DN Gs and Nb35. It is known from class B GPCR cryo-EM structures that the ECD of these receptors is mobile and capable of adopting more than one conformation in a peptide bound state. It is highly likely that the authors have captured two states with the peptide bound, given ligand was present in the prep, not an unbound and bound state, however without higher resolution cryo-EM data, this cannot be absolutely determined. If the authors speculate the GLP-1R can adopt a G protein bound state in the absence of peptide, then they should be able to trap this complex to image by negative stain. To confidently compare conformational movements within the ECD by neg stain, the authors require 3 different preps, not one (1. The apo receptor – no peptide, no G protein or alternatively their crystallisation prep; 2. Receptor Gs no peptide and 3. The sample used in Fig 4).

Page 3: Introduction: It is unclear what the authors mean by "previous inactive GLP-1R TMD structures revealed an allosteric regulation mechanism". What mechanism was this and how did the inactive TMD structures reveal this. I am not convinced these structures revealed any allosteric mechanism at all, so this needs to be clarified.

Page 3: the authors refer to the pocket volume – presumably this refers just to the TMD not the full peptide binding site?

The disulphide experiments are very elegant and provide convincing evidence for interactions between the TMD and ECD. Were any of the receptor expression levels changed between the mutant and WT receptors?

The authors show comparisons and discuss the GLP-1R inactive state solved in this paper to an earlier structure of the full length GCGR inactive state. While I think this is important to include, the authors need to be careful with the way this is described. The GCGR inactive conformation in the published x ray structure does not represent the same inactive conformation solved here of the GLP-1R. It should be noted up front that the GCGR inactive structure is bound by an antibody that acts as an antagonist of GCGR binding. As such, the ECD in this structure has to be released as this antibody binds in the peptide binding groove. While many of the comparisons between the two receptors are valid

(differences in stalk and TM2-ECL1-TM3 conformations), the ECD comparisons are not directly valid as they represent two different states, due to the mechanism by which they were stabilised. This should be highlighted in more detail. In addition, in the conclusion statement, the section on the GCGR should be revised to reflect this (as GCGR, they have not trapped something resembling the apo state).

In the discussion – ref 23 is referenced when discussing ECL3 mutations. This reference is for ECL3 mutations in the CLR, not GLP-1R (as read it implies it is the GLP-1R). ECL3 mutations have been studied for the GLP-1R and there is no evidence of constitutive activity for this receptor from single ala mutations. This section should be reworded to address this. ie: I think what the authors are trying to say is that the closed conformation likely exists for all class B GPCRs with ECD-ECL3 interactions stabilising the inactive state and this is supported by ECL3 mutagenesis studies where they are using this reference as an example. This section of the discussion should be reworded to reflect this.

On page 10 in the discussion it is unclear to me what is meant by “antagonism feature of the ECD represents an alternative mechanism of activation”. This new structure and the role of the ECD revealed here, does not alter my understanding of class B GPCR activation, but finally provides solid experimental validation that supports the current theory for class B GPCR activation by peptides that initially requires release of the ECD from the TMD. If the authors are speculating a new mechanism of activation, then they should provide details of what this is and how this differs from the current understanding.

The very last statement states “this structure may prove crucial in design of allosteric modulators”. Why specifically allosteric modulators? and how would this structure be used for this. Allosteric modulators for this receptor class have been extremely difficult to identify and there are very few compounds reported. Does this structure identify novel druggable pockets? This is not clear to me how this structure would be used for designing allosteric modulators.

Methods

This structure contained an additional mutation relative to the one used previously. Why was this additional mutation necessary? Is this construct actually capable of binding peptide agonists?

cAMP studies: These were performed using a FLAG tagged receptor. Please provide details of the construct and its pharmacology. Were the glosensor experiments performed in the absence or presence of IBMX?

Fab7F38-Semaglutide-GLP-1R-Gs: Please provide details of the receptor construct and the DNGs construct used here (tags and their positions, mutations and pharmacology). Please also provide the size exclusion profiles of the complexes.

Supplemental Figure legend 3: “red circles indicate crashed area when GLP-1 is docked into the GLP-1R inactive structure”. This should be reworded to state “red circles indicate where clashes occur when GLP-1 is docked into the inactive structure”.

Reviewer #3:

Remarks to the Author:

The authors report the 3.2 Å resolution, peptide-free crystal structure of the full-length human GLP-1R in an inactive state that reveals a unique closed conformation of the ECD. The authors proposed a biochemical validation for the physiological relevance of the closed conformation while electron microscopy (EM) and molecular dynamic (MD) simulations suggest a large degree of conformational dynamics is associated with the ECD and is necessary for binding to GLP-1. There is a clear demonstration of the close conformation of the orthosteric binding pocket by the ECD domain. The structure presented here highlight mainly differences around EC loops and ECD compared to previously released structure of GLP-1R bound to agonist or allosteric inhibitors, as well as, closely related GCGR receptor .

This study is very focussed and represents a detailed analysis of GLP-1R ECD domain orientation related to 7TM domain in inactive conformation. I am not convinced by EM negative stain analysis

and conclusions drawn by the authors. The resolution is very low, the authors do not know the ligand occupancy of the receptor complexes. Additionally they show that the Fab7F38 does not prevent receptor activation and does not affect ECD dynamics. It is highly speculative and data are over interpreted in my opinion, to finally conclude that « This indicates that the intracellular part of GLP-1R can assume an active state coupling to G_s, while the ECD adopts a closed inactive conformation without agonist binding ». In the mean time, this can be possible since this study does not show any role of ECD on reducing basal/constitutive activity of the GLP-1R associated with the close conformation reported here. The close conformation is more likely to prevent access of orthosteric binding site to GLP-1, as presented here. However, the discussion suggest that « Complementarily, a previous study showed that point mutations in ECL3 lead to significant increases in both basal and ligand-induced activity of a class B GPCR, potentially through the breakdown of the ECD–ECL3 interactions by these mutations that further relieves TMD from a ECD induced autoinhibition state. ». As a consequence, The study presented here does not go along with a potential role of ECD in controlling basal activity, which makes the role of ECD and the presented conformation, confusing. Is the ECD antagonising the receptor or is this conformation compatible with G protein binding? In that respect, despite of a possible biologically relevant conformation, function of such conformational state does not seem clear cut and needs clarification.

An interesting point is the volume of the orthosteric-binding site that remains even, upon activation. This is rather interesting considering that for class A, contraction of the ligand-binding site was previously reported as a first step toward receptor activation. What is the volume of orthosteric site for the inactive conformation previously published?

Overall this article is of interest for class B GPCRs community but could also read better for broad readership .

NCOMMS-19-7936194
Point-by-point responses

Reviewer 1

This is a very interesting paper revealing new features regarding the mechanism of action of Family B GPCRs.

Response: We thank the reviewer's positive comment on our work.

One minor point to address.

While I understand what the authors are trying to convey, the statements on p10 referring to the ECD being an "antagonist" do not make pharmacological sense.

"...the receptor is antagonized in its apo-state. The antagonism feature of the ECD revealed here indicates..."

An antagonist is defined as a ligand that reduces the action of another ligand (an agonist), which is clearly not the case in the situation with the apo-state GLP-1R. The language needs to be modified to avoid misusing this pharmacological term.

Response: We appreciate the comment. By saying "antagonized" or "antagonism feature" we meant the receptor is negatively regulated in its apo-state by the ECD. We agree that the descriptions are not accurate, and we have thus removed them in revised manuscript and used "negatively regulated" instead.

"... The inactive structure aligns with the dual functional hypothesis of the ECD through which, on one hand, peptide binding and activation is mediated when the ligand is present, while on the other, the receptor is negatively regulated in its apo-state".

Reviewer 2

This paper reports a novel crystal structure of the GLP-1R in a peptide free state that likely represents the inactive conformation of the receptor. It reveals a unique conformation of the ECD that is closed over the TM bundle. This structure is also supported by crosslinking data confirming specific interactions within the ECD and ECLs that stabilise the closed conformation. They also use a series of MD experiments to propose a model of class B GPCR activation by peptides and use neg stain EM, MD and previously published structures to provide a model for class B GPCR activation by peptide agonists. While this paper itself does not change our current understanding of class B GPCR activation by peptide agonists, it does solidify our understanding. A closed conformation of the receptor ECD in the apo state has been predicted for some time and is consistent with earlier MD studies on the GCGR and GLP-1R and mutagenesis studies on the SecR and GCGR that supported the notion that the inactive state is stabilised by a closed ECD conformation mediated by ECD-ECL3 interactions that need to be released for peptide engagement. This is the first time that this closed conformation of the receptor has been experimentally validated and visualised, revealing not only extensive interactions between the ECD and ECL3, but also with ECL1 and therefore represents a significant advance for the class B GPCR field. I have a number of comments on the manuscript in its current form.

Response: We thank the reviewer for this positive comment.

Main comments

In the title and abstract, this structure is referred to as an apo-GLP-1R. While the authors speculate that the structure represents an apo state, the structure is bound to both a NAM and an antibody and is therefore by definition not an apo structure. In addition, the receptor has been stabilised in an inactive conformation (by both the use of a NAM and mutagenesis) and therefore is unlikely to sample the conformational landscapes occupied by the apo receptor. This should therefore be revised to state either inactive or peptide-free, not apo.

Response: We thank the reviewer for the comment and acknowledge that the receptor is not in an absolute apo-state, and therefore, we have changed the term to “peptide-free” throughout the manuscript including a change in the title.

One caveat to this study is that the receptor contains 11 thermostabilising mutations including a disulphide stabilising TM5-6 and mutations at the top of TM7 that may influence the conformation, particularly the location of ECL3, however given the authors have shown convincing data supporting the interactions in the x ray structure, this is not too much of a concern. Did the authors try generating crystals with a construct containing less mutations?

Response: We generated several 7TM constructs with fewer mutations and found that back mutation of two disulfide-unlinked cysteines (C193S, C233M) did not affect the crystal quality, while back mutations of other residues (e.g., F196I, A271S) generated inferior crystals. Related results have been described in a recent publication [Xu, et al. (2019). *IUCrJ.* 6, 996–1006]. We applied all 10 thermostabilizing mutations and E387D (a mutation to mimic the identified interaction in the GCGR-NNC1702 complex structure) in the full-length construct (we tried co-crystallization with the peptide analogue but were not successful).

In this structure PF06372222 was used as a NAM to stabilise the structure. It should be clearly stated (currently it is not mentioned anywhere) that this is not a NAM of the native GLP-1R (it is a GCGR NAM) and only behaves as a NAM in the modified construct used in this study that contains residues from the GCGR that allow this to bind the GLP-1R. The authors have used this construct and NAM previously, and there is some confusion in the literature where this compound has been reported as a GLP-1R NAM, where it actually isn't. If the authors are using a tool compound that only behaves at a modified non-native receptor, it is their duty to make this clear in the paper.

Response: We appreciate this comment. The NAM has been used in our previous GLP-1R 7TM paper [Song, et al. (2017) *Nature.* 546, 312-315] where we stated “these NAMs were previously optimized for GCGR antagonism, but certain analogs were found to antagonize GLP-1R as well”. In the current manuscript, we did introduce a mutation to mimic GCGR and further improve the antagonistic feature of PF06372222. This was necessary for the crystallization of both 7TM and full-length GLP-1R. To clarify this in the text, we have now added the reference PF-06372222

and a note showing that this was originally designed for GCGR.

In figure 4 (and the related text) the authors show low resolution negative stain EM data of semaglutide bound the GLP-1R:Gs complex. Here they speculate that they have two different states, one bound by semaglutide and one unliganded and they have therefore docked their inactive model into one of these classes even though the class average clearly contains G protein). They speculate that the GLP-1R can bind Gs in the absence of ligand. This section is highly speculative and is not supported by any experimental data. It is not appropriate to state the GLP-1R can adopt an intracellular active conformation capable of coupling to G protein without agonist peptide binding, when there is absolutely no data presented to support this. While it is possible, it is unlikely to be the case here. The complex has been stabilised using a high concentration of semaglutide, DN Gs and Nb35. It is known from class B GPCR cryo-EM structures that the ECD of these receptors is mobile and capable of adopting more than one conformation in a peptide bound state. It highly likely that the authors have captured two states with the peptide bound, given ligand was present in the prep, not an unbound and bound state, however without higher resolution cryo-EM data, this cannot be absolutely determined. If the authors speculate the GLP-1R can adopt a G protein bound state in the absence of peptide, then they should be able to trap this complex to image by negative stain. To confidently compare conformational movements within the ECD by neg stain, the authors require 3 different preps, not one (1. The apo receptor – no peptide, no G protein or alternatively their crystallisation prep; 2. Receptor Gs no peptide and 3. The sample used in Fig 4).

Response: We appreciate this valuable comment and agree that the receptor most likely adopts multiple conformations in the presence of peptide agonist because of the dynamics of ECD (e.g., the PTH1R and CTR EM structures). We recognize that our previous analysis of the EM data was too speculative and have decided to focus only on the 2D class averages which provide clear evidence for multiple ECD conformations, some of which closely resemble the active state determined previously by cryo EM, while other 2D classes clearly differ from the active state. However, as pointed out by the reviewer, due to the limited resolution of the EM data it is not possible to conclude about the molecular details of the various conformations. We will continue to work on improving the resolution and data quality but this may take a significant amount of time and we feel the current data is timely for the field and has value. We thank the reviewer for suggesting further experiments, but believe that the current EM results are sufficient to demonstrate the flexibility of ECD, the non-competitive feature of Fab7F38, and suggest that Fab7F38 does not interfere with the conformational flexibility of the ECD during crystallization. Therefore, we have revised the manuscript accordingly throughout the text and figures (Fig. 4 and supplementary Fig. 5):

EM analysis of Fab7F38-bound GLP-1R

The ECD of agonist-bound GLP-1R is known to assume different orientations in a ligand-dependent manner^{3,4,12}. Conformational flexibility of the ECD was also observed in the Fab7F38-bound GLP-1R using negative stain EM single-particle analysis of a complex consisting of Fab7F38, semaglutide (a closely related analogue of GLP-1 and approved drug for treatment of type 2 diabetes), detergent solubilized GLP-1R, Gs (nucleotide free), and Nb35 (Fig. 4, Supplementary Fig. 5). The 2D class averages clearly show Fab7F38 bound to the ECD and

the TMD in detergent micelle with the associated Gs protein (stabilized by Nb35). Interestingly, the 2D class averages reveal multiple conformations of the Fab7F38-bound ECD with some conformations closely resembling an open conformation as observed in the fully active structures (Fig. 4a). The ECD was reported to be relatively dynamic even in the presence of hormone peptide. In fact, in an analogous class B receptor, the parathyroid hormone receptor-1 (PTH1R), the ECD can adopt more than one conformation while bound to a long-acting PTH analogue and in the Gs-coupled state¹⁹. Similarly, a preparation of calcitonin receptor (CTR)-calcitonin-Gs complex did not resolve clear density for the ECD of CTR attributed to partial flexibility⁸. Fig. 4b shows a model of Fab7F38-bound active state GLP-1–GLP-1R–Gs superimposed on a pair of 2D class averages, suggesting that GLP-1R can assume an open active conformation while bound to Fab7F38. In contrast, the active model does not align with the tilted 2D classes (Supplementary Fig. 5c). These conformations indicate flexibility of the ECD even in the Gs-bound state, however, the limited resolution of the EM data does not allow for a precise conclusion about the molecular details of the various conformations.

Page 3: Introduction: It is unclear what the authors mean by “previous inactive GLP-1R TMD structures revealed an allosteric regulation mechanism”. What mechanism was this and how did the inactive TMD structures reveal this. I am not convinced these structures revealed any allosteric mechanism at all, so this needs to be clarified.

Response: Our previous paper [Song, et al. (2017) Nature. 546, 312-315] reported that the TMD structures did indicate the mechanism of how GLP-1R may be allosterically regulated by small molecules and potential directions for drug development. In this previous work, we crystallized the 7TM in complex with two NAMs and built a model of the receptor in complex with a positive allosteric modulator (compound 2). The model of compound 2 was verified by the mutagenesis data, which showed that NAMs locate outside of helices V-VII thus blocking the outward shift of helix VI for downstream signaling, while compound 2 targets the same general region but mainly interacts with helices V and VI, which may facilitate the formation of an intracellular binding site that enhances G protein coupling.

In view of this comment, we have modified our description: “*Previous inactive GLP-1R TMD structures revealed how allosteric modulators can precisely regulate its function from the extra-helical binding sites....*”.

Page 3: the authors refer to the pocket volume – presumably this refers just to the TMD not the full peptide binding site?

Response: Yes, we measured the volumes of the TMDs and have clarified in the revised text.

The disulphide experiments are very elegant and provide convincing evidence for interactions between the TMD and ECD. Were any of the receptor expression levels changed between the mutant and WT receptors?

Response: The expression levels of mutants are about 90%-100% of the wild-type which is now

described in the revised method section.

The authors show comparisons and discuss the GLP-1R inactive state solved in this paper to an earlier structure of the full length GCGR inactive state. While I think this is important to include, the authors need to be careful with the way this is described. The GCGR inactive conformation in the published x ray structure does not represent the same inactive conformation solved here of the GLP-1R. It should be noted up front that the GCGR inactive structure is bound by an antibody that acts as an antagonist of GCGR binding. As such, the ECD in this structure has to be released as this antibody binds in the peptide binding groove. While many of the comparisons between the two receptors are valid (differences in stalk and TM2-ECL1-TM3 conformations), the ECD comparisons are not directly valid as they represent two different states, due to the mechanism by which they were stabilised. This should be highlighted in more detail. In addition, in the conclusion statement, the section on the GCGR should be revised to reflect this (as GCGR, they have not trapped something resembling the apo state).

Response: We appreciate the comment related to the comparison of GLP-1R and GCGR and have revised the manuscript accordingly:

“The Fab7F38-bound GLP-1R is crystallized in a closed conformation with the ECD’s peptide binding area sealed, in line with the non-competitive property of Fab7F38. The ECD of GCGR on the other hand is in an extended conformation with the β -sheet module covering the orthosteric TMD pocket and the inhibitory mAb1 engaging the orthosteric ECD binding site directly. Clearly, mAb1-bound GCGR is unable to assume the closed conformation observed in the Fab7F38–GLP-1R structure.”

And furthermore in the section of the manuscript comparing GLP-1R and GCGR:

“Different types of antibodies facilitated crystallization of GLP-1R and GCGR into varied ECD orientations, whereas the structural difference in the TMD extracellular regions, including the stalk of helix I, is potentially due to sequence diversity between the two receptors”.

In the discussion – ref 23 is referenced when discussing ECL3 mutations. This reference is for ECL3 mutations in the CLR, not GLP-1R (as read it implies it is the GLP-1R). ECL3 mutations have been studied for the GLP-1R and there is no evidence of constitutive activity for this receptor from single ala mutations. This section should be reworded to address this. ie: I think what the authors are trying to say is that the closed conformation likely exists for all class B GPCRs with ECD-ECL3 interactions stabilising the inactive state and this is supported by ECL3 mutagenesis studies where they are using this reference as an example. This section of the discussion should be reworded to reflect this.

Response: We appreciate this suggestion and recognize that the increased basal activity by ECL1/ECL3 mutations is found in GCGR (this is now clearly stated in the revised version). The GLP-1R closed conformation is supported by our disulfide trapping experiments, and we intended to indicate that the closed conformation likely exists for other class B GPCRs and conformational changes are needed to fit the peptide ligand in the canonical pathway. We have now rephrased the relevant text accordingly:

“The ECD has been proposed to function as a negative regulator of GCGR. Specifically,

mutagenesis of ECD, ECL1 and ECL3 individually increased the basal activity of GCGR, which could result from the interruption of intramolecular interactions between the ECD and ECL1/3 that otherwise stabilize the receptor in an inactive state. The inactive structure of GLP-1R presented here aligns nicely with a possible role of the GLP-1R ECD as a negative regulator of GLP-1R, although, in contrast to GCGR, we and others have failed to identify mutations in the ECD/ECL interface that increase basal activity of GLP-1R. However, this could reflect differences in the TMD of GLP-1R and GCGR and their intrinsic ability to activate G protein in the absence of agonist, independent of the ECD”.

Our structure can explain some but not all previous data. Therefore, in the revised version we mention that it is possible to have non-canonical pathways and suggest the ECD could play a different role assuming different conformations depending on the ligand. We have reorganized the relevant paragraph and believe that the revision may answer this as well as the question below (see below).

On page 10 in the discussion it is unclear to me what is meant by “antagonism feature of the ECD represents an alternative mechanism of activation”. This new structure and the role of the ECD revealed here, does not alter my understanding of class B GPCR activation, but finally provides solid experimental validation that supports the current theory for class B GPCR activation by peptides that initially requires release of the ECD from the TMD. If the authors are speculating a new mechanism of activation, then they should provide details of what this is and how this differs from the current understanding.

Response: We agree that our previous description of “antagonism feature of the ECD represents an alternative mechanism of activation” is misleading. What we meant was, in addition to the canonical orthosteric peptide induced pathway, there may exist some non-canonical mechanisms that do not need the peptide N-terminus to dock into the pocket, and the triggers of TMD conformational changes could be other type of ligands, e.g., the non-peptide agonist, TT-OAD2. In the non-canonical mechanisms of class B receptors, the ECD could play a different role assuming different conformations depending on the ligand. Below are the revisions we made: *“According to the model presented here based on current inactive and active structures of GLP-1R, the canonical peptide activation pathway involves a major conformational change of the ECD, going from closed inactive to open active state (Figure 7). Non-canonical activation mechanisms may exist and the ECD could play a different role assuming different conformations depending on the ligand. Most recently, in a new structure of GLP-1R, a small molecule agonist, TT-OAD2, was shown to activate the receptor through binding of the helix II–ECL1–helix III region, revealing another ECD conformation (Nature, in press).”*

The very last statement states “this structure may prove crucial in design of allosteric modulators”. Why specifically allosteric modulators? and how would this structure be used for this. Allosteric modulators for this receptor class have been extremely difficult to identify and there are very few compounds reported. Does this structure identify novel druggable pockets? This is not clear to me how this structure would be used for designing allosteric modulators

Response: By saying “allosteric modulators” we intended to raise the possibility of inspiring the

design of new molecules targeting the ECD–ECL interface. For example, a small molecule that can help to break ECD-TMD interactions may have similar effects as the ECL1/3 mutations, thus could be a positive allosteric modulator of GLP-1. We rephrased the sentence to clarify: *“The new structure enables a comparison of the only two inactive peptide-free class B GPCR structures and increases our structural understanding of their activation pathways, which may be helpful in the design of novel ligands, thereby enabling new avenues for drug discovery on therapeutically important class B GPCRs.”*

Methods

This structure contained an additional mutation relative to the one used previously. Why was this additional mutation necessary? Is this construct actually capable of binding peptide agonists?

Response: The aim of introducing the mutation of E387D was to increase binding with a GLP-1 antagonist analogue (to mimic the identified interaction in the GCGR-NNC1702 complex structure). We previously wanted to simultaneously determine the complex structure of current construct with the peptide analogue but were not successful. This was expected since binding of peptide to this crystallization construct might be affected because of the mutations introduced in the TMD that lock the protein into inactive conformation (although the ECD is still intact). Glu and Asp are quite similar residues, so the mutation does not change too much which is not located in the ECD-TMD interface.

cAMP studies: These were performed using a FLAG tagged receptor. Please provide details of the construct and its pharmacology. Were the glosensor experiments performed in the absence or presence of IBMX?

Response: We have now provided construct details and its pharmacology in the revised manuscript. The experiment was performed in the absence of IBMX.

Fab7F38-Semaglutide-GLP-1R-Gs: Please provide details of the receptor construct and the DNGs construct used here (tags and their positions, mutations and pharmacology). Please also provide the size exclusion profiles of the complexes.

Response: Detailed information has been updated in the revised manuscript as requested.

Supplemental Figure legend 3: “red circles indicate crashed area when GLP-1 is docked into the GLP-1R inactive structure”. This should be reworded to state “red circles indicate where clashes occur when GLP-1 is docked into the inactive structure”.

Response: We have now reworded this in the revised manuscript.

Reviewer 3

The authors report the 3.2 Å resolution, peptide-free crystal structure of the full-length human GLP-1R in an inactive state that reveals a unique closed conformation of the ECD. The authors

proposed a biochemical validation for the physiological relevance of the closed conformation while electron microscopy (EM) and molecular dynamic (MD) simulations suggest a large degree of conformational dynamics is associated with the ECD and is necessary for binding to GLP-1. There is a clear demonstration of the close conformation of the orthosteric binding pocket by the ECD domain. The structure presented here highlight mainly differences around EC loops and ECD compared to previously released structure of GLP-1R bound to agonist or allosteric inhibitors, as well as, closely related GCGR receptor.

This study is very focused and represents a detailed analysis of GLP-1R ECD domain orientation related to 7TM domain in inactive conformation. I am not convinced by EM negative stain analysis and conclusions drawn by the authors. The resolution is very low, the authors do not know the ligand occupancy of the receptor complexes. Additionally they show that the Fab7F38 does not prevent receptor activation and does not affect ECD dynamics. It is highly speculative and data are over interpreted in my opinion, to finally conclude that « This indicates that the intracellular part of GLP-1R can assume an active state coupling to Gs, while the ECD adopts a closed inactive conformation without agonist binding ». In the mean time, this can be possible since this study does not show any role of ECD on reducing basal/constitutive activity of the GLP-1R associated with the close conformation reported here.

Response: We thank the reviewer for this important comment. In terms of the EM data, we recognize that our previous analysis on EM might be overly speculative. This issue was also raised by Reviewer 2 and significant changes of the manuscript have been implemented accordingly, which we believe also address the concerns by Reviewer 3 (see above).

We agree that the different 2D classes most likely represent different agonist-bound ECD conformations, as pointed out by reviewer 2, however we feel it is not appropriate to ignore the fact that the inactive crystal structure aligns nicely with some of the observed 2D classes and therefore we speculate about the less likely possibility of a GLP-1R/Gs complex in the absence of agonist in a fraction of the purified receptor complex (in the revised discussion):

“The multiple conformations observed from the single particle EM analysis of the purified Fab7F38–semaglutide–GLP-1R–Gs–Nb35 complex most likely represent different agonist-bound ECD conformations, however some 2D classes are also compatible with the closed conformation observed in our inactive crystal structure (Supplementary Fig. 5d). We speculate that a fraction of the purified receptor complex may bind Gs in the absence of the agonist. Although less likely, such a scenario is not unusual, and has been visualized in the prototypical β 2 adrenergic receptor: both ligand-free and antagonist-bound β 2 adrenergic receptors can form a complex with nucleotide-free Gs^{23,24}. The non-competitive Fab7F38 could be an important tool for further structural clarification of the existence of different states of GLP-1R.”

The close conformation is more likely to prevent access of orthosteric binding site to GLP-1, as presented here. However, the discussion suggest that « Complementarily, a previous study showed that point mutations in ECL3 lead to significant increases in both basal and ligand-induced activity of a class B GPCR, potentially through the breakdown of the ECD–ECL3 interactions by these mutations that further relieves TMD from a ECD induced autoinhibition state. ». As a consequence, the study presented here does not go along with a potential role of

ECD in controlling basal activity, which makes the role of ECD and the presented conformation, confusing. Is the ECD antagonising the receptor or is this conformation compatible with G protein binding? In that respect, despite of a possible biologically relevant conformation, function of such conformational state does not seem clear cut and needs clarification.

Response: The negative regulation of receptor function by ECD was reported (Ref 21 in the text) previously on GCGR. Our structure aligns nicely with a dual function of the GLP-1R ECD through which, on one hand, peptide binding and activation is mediated when the ligand is present, while on the other, the receptor is negatively regulated in its apo-state. Clearly, the closed conformation has a “negative” impact by preventing GLP-1 binding as kindly pointed out by Reviewer 3 and we have emphasized that aspect in the revised discussion. Accordingly, GLP-1 induced activation is significantly impacted by locking the ECD onto the TMD by disulfide bridges. In contrast, it is less clear to what extent the closed conformation impacts Gs binding (precoupling, not Gs activation) or basal activity of GLP-1R. We acknowledge that further experiments are necessary to fully understand the functional impact of the ECD conformations. Thus, we have edited the text accordingly:

“The ECD has been proposed to function as a negative regulator of GCGR. Specifically, mutagenesis of ECD, ECL1 and ECL3 individually increased the basal activity of GCGR, which could result from the interruption of intramolecular interactions between the ECD and ECL1/3 that otherwise stabilize the receptor in an inactive state. The inactive structure of GLP-1R presented here aligns nicely with a possible role of the GLP-1R ECD as a negative regulator of GLP-1R, although, in contrast to GCGR, we and others have failed to identify mutations in the ECD/ECL interface that increase basal activity of GLP-1R. However, this could reflect differences in the TMD of GLP-1R and GCGR and their intrinsic ability to activate G protein in the absence of agonist, independent of the ECD. Obviously, the ECD of the closed conformation prevents GLP-1 binding and further experiments are necessary to fully understand the functional impact of the closed ECD conformation on Gs binding.”

An interesting point is the volume of the orthosteric-binding site that remains even, upon activation. This is rather interesting considering that for class A, contraction of the ligand-binding site was previously reported as a first step toward receptor activation. What is the volume of orthosteric site for the inactive conformation previously published?

Response: It is correct that for class B receptors the volume of TMD does not change much before and after peptide binding. This was quite different from class A GPCRs. We recalculated the volume for previous inactive structure to 1350 Å³, which is larger than the others (893-1036 Å³). However, we do not think this number is reliable as the ECL1 and ECL3 loops are missing in the previous structure so the volume is apparently overestimated (thus not added to the text). In contrast, all extracellular loops are completely modeled with the other three structures. Furthermore, the current inactive structure is quite similar as that previous TMD (C α r.m.s.d: 0.6 Å).

Reviewers' Comments:

Reviewer #2:

Remarks to the Author:

The authors have addressed all of my comments in their revised manuscript.

Reviewer #3:

Remarks to the Author:

The authors addressed all my comments.